# Risk perception and coping response to COVID-19 mediated by positive and negative emotions: A study on Chinese college students

**Yongtao Gan**[1], **Qionglin Fu**[2]*

**1** College of law, Shantou University, Shantou City, China, **2** College of Education, Hubei University, Wuhan City, China

* catganwithall@163.com

**Data Availability Statement:** The data underlying the results presented in the study are available from: https://www.protocols.io/view/risk-perception-and-coping-response-to-covid-19-is-bzzfp73n/metadata.

## Abstract

This study aimed to assess the mediating roles of positive and negative emotions on the relationship between COVID-19-related risk perception and coping behaviours adopted by Chinese college students in response to the COVID-19 pandemic. We conducted an internet-based questionnaire survey from mid February–late October 2020, among 1038 college students, from six Chinese universities (females = 73.41%), ranging within 17–26 years. The survey questionnaire included three major components—the COVID-19-Related Risk Perception Scale (CRPS), the Positive and Negative Affect Scale (PANAS-Revision), and Coping Response of COVID-19 Scale (CRCS). Descriptive statistics and a mediated model were used to analyse the collected data. A partial mediation relationship was found between COVID-19-related risk perception and 1) active-response behaviour (β = 0.05, 95% Confidence Interval [CI: 0.03, 0.08]), 2) self-protection behaviour through positive emotions (β = 0.03, CI [0.01, 0.04]), and 3) risk-taking behaviour through negative emotions (β = -0.04, CI [-0.07, -0.02]). This study's double-mediation model has been shown to detect the effect coping mechanisms to COVID-19. Furthermore, it implies that public health managers should consider the differences in coping mechanisms and the diverse mediating roles of positive and negative emotions for coping with public health emergencies.

## Introduction

The coronavirus disease 2019 (COVID-19) was first detected on December 2019 in Wuhan, the Hubei Province of China [1]. Patients with COVID-19 developed fever, cough, breathlessness, diarrhoea, and fatigue. Initially, patients died from complications of renal insufficiency, pneumonia, and severe respiratory problems [2]. COVID-19 was declared a Public Health Emergency of International Concern (PHEIC) in late January 2020. This declaration led to stringent restrictions worldwide and indicated serious consequences like the global spread of the disease or infection [3].

**Funding:** This study was funded by the National Social Science (Educational Project) Foundation of China, Grant/Award Number: BMA18004. Funding body had no influence on study design, data collection, data analysis, data interpretation or writing the manuscript.

**Competing interests:** The authors declare no conflict of interest.

There is an expansive body of research on the public's coping response and risk perception [4,5]. In general, people tend to judge risk by their perception of risk [3], intuitively evaluating the danger faced by them [6], including potential negative consequences they may associate with it [7]. Surveys on how people accept and respond to public health risks were proposed by Slovic [8], holding the departments responsible for assessing and managing public health and safety needs. Researchers explored the explanations underlying the cultural and social perspectives and the risk reality in the sociological field; however, they explored the individual factors associated with the risk perception and its assessment with a cognitive perspective in the psychological field [9]. Recently, researchers from both lines of enquiry came together to investigate these issues at a deeper level. The groundwork for studies exploring the public response has been laid by the research regarding risk perception. Shi et al. [10] observed that a positive coping behaviour positively affected risk perception. Therefore, to prevent the public from acting irrationally, the risk perception needs to be maintained at a reasonable level, as noted by Drace [11]. In a study programme in China, Wang et al. [12] observed that the attitudinal awareness, knowledge, and behaviour regarding emergencies could be strengthened through participatory training on emergency response preparedness.

Billings and Moos [13] divided the coping methods into active behavioural, active cognitive, and avoidance. Nonetheless, earlier research observing local residents and COVID-19 characteristics found that the coping responses to COVID-19 were segregated into: 1) active-response behaviour (ARB), involving measures for reducing the possibility of risk occurrence and their harm; 2) self-protection behaviour (SPB), that is, reducing COVID-19 risk through a healthy diet, green food, and physical exercise; and 3) risk-taking behaviour (RTB), that is, ignoring the risky situation one is in and showing behaviours that may threaten one's safety.

Moreover, people tend to feel threatened by the disease, resulting in added stress, anxiety, and depressive symptoms, which lead to further psychological problems and emotional drainage [14,15]. Emotions could be divided into negative and positive from the perspective of valence. The physiological changes observed while experiencing various emotions have been the subject matter of recent studies. Systematic literature reviews in countries such as China, Iran, Italy, and Nepal suggest that depression, anxiety, and stress presented themselves in various elevated levels during the pandemic? Fear of COVID-19 is widespread and experienced within different populations [16–20]. In one specific study, under various situations like being happy, distressed, or being emotionally neutral, the diameter of infants' pupils was measured. The authors found that both negative emotions (NE) and positive emotions (PE) were displayed in 6–12-month-old babies [21]. Recent studies have shown that prolonged running accentuated the PE and dampened the NE in the prefrontal cortex, though it could not be considered an ultimate success [22].

In the human body, the change of emotions could directly affect the different behavioural and physiological responses, as revealed by the results of medical research [23,24]. People's cognitive abilities are expanded by PE resulting in enhanced intellectual, social, and physical resources. The overall health and quality of life of an individual could be highly improved by reducing their stress or pressure by instilling PE. In contrast, NE can affect the individual's physiological functioning, interfere with the immune system, and induce various psychosomatic and mental illnesses. Increased fear levels toward COVID-19 were associated with decreased job satisfaction, prevalence of sleep problems, increased psychological distress, and increased organizational and professional turnover intentions [16,17].

During COVID-19, college students face various major life challenges such as decreased efficiency of study time and the effort expended by studying, and they are under constant pressure. If they cannot adapt to the changes brought by these life events, they will be prone to

anxiety, loneliness, feelings of worthlessness, and other NE. If these resulting NE are not curbed, they will seriously affect the physical and mental health of the students.

Numerous studies have explored the emotional activities involved in the negative and positive emotional responses relating to social behaviours or psychological symptoms [21]. Cho et al. [25] explained that schizophrenic patients indicated similar experiences to stimuli that elicit PE or NE. In terms of social behaviour, social anxiety affects the strategy of emotional regulation in terms of its result, type, and frequency [26]. Recent studies have indicated the coexistence of negative and PE of self-consciousness related to the risk of transmission after HIV diagnosis [27].

The more college students are willing to reduce the pressure or threat of COVID-19 through their behaviours, the higher their risk perception. In contrast, those who do not think COVID-19 has risks, such as college students with low COVID-related risk perception (CRP), may ignore their situation and attempt to live and work as normal [28].

Although previous studies have paid attention to risk perception in education institutions, limited research has been conducted on the impact of CRP among student members in universities, or the mediating roles of PE and NE. Hence, this study aims to determine how NE and PE mediate the relationship between CRP and the coping responses to COVID-19 (CRC) of students in Chinese colleges.

Therefore, in accordance with previous research, CRP may also predict PE, NE, and CRC. Thus, this study hypothesized that PE and NE would mediate the associations between CRP and three categories of CRC (i.e. ARB, SPB, and ATB) among Chinese college students.

In the light of the literature reviews and hypothesis, this study presented a theoretical model, which has been presented in Fig 1.

## Materials and methods

This descriptive correlational study quantitatively examined how PE and NE mediated the relationship between CRP and CRC among college students in China. The survey was

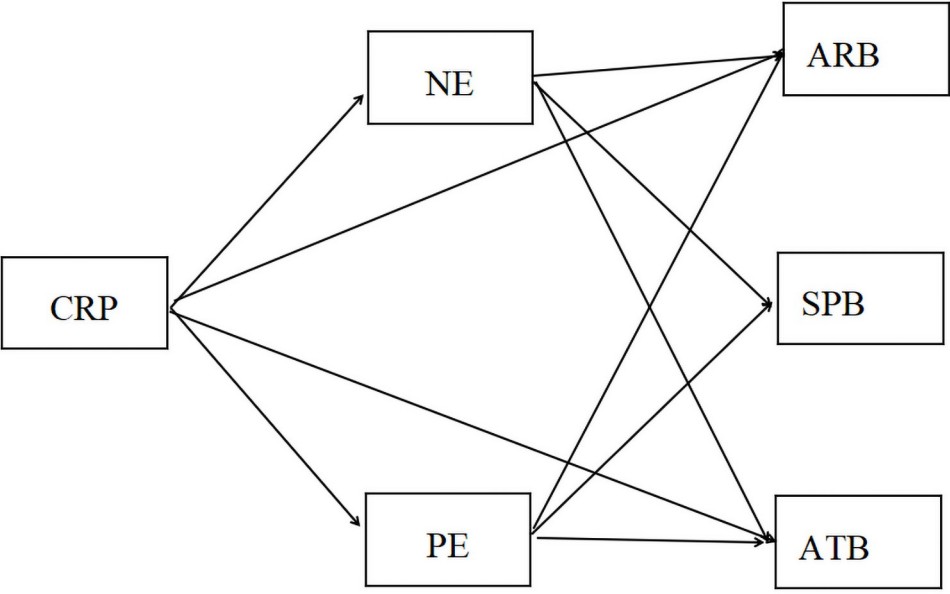

**Fig 1. Concept model.** SPB = Self-protection behaviour, PE = positive emotion, NE = negative emotion, ARB = Active-response behaviour, SPB = Self-protection behaviour, CRP = COVID-19-related risk perception, RTB = Risk-taking behaviour.

conducted with six universities in three provinces (Guangdong, Hubei, and Shandong) of China. All participants completed and submitted the questionnaires during the data collection period (17 February to 10 March 2020).

## Data collection

First, the universities in the Hubei, Guangdong, and Shandong Provinces were sent an invitation e-mail with the study purpose information and a link to an online survey. Six universities responded and agreed to enrol their students in the study. The questionnaire covered the following domains: sociodemographic details, CRC, negative and positive emotions, and self-reported CRP. Subsequently, the link, www.wenjuan.com was disseminated by our colleagues to all the eligible college students through a QQ group and a WeChat group—two of the most widely used social media platforms in China. To ensure valid results, we imposed the following inclusion criteria for partipants: 1) They should be currently studying in universities and 2) Participants could only respond to the questionnaire once to avoid repetition/redundant data. The questionnaire was modified accordingly after conducting a pilot study on 09 February using 30 respondents. Later, during the data collection period, the participants completed and submitted the formal questionnaires (17 February to 10 March 2020). During this period, all higher education institutions were closed for 5–7 weeks. Before responding to the questionnaire, all participants provided written informed consent form to gain access to the research process. Of the 1322 completed questionnaires, those with too many default values or missing values were excluded from the analysis; consequently, 1038 valid responses were considered.

## Research ethics

This study was approved by Research Ethics Committee of the Shantou University medical college (SUMC-2020-80). Participants provided written informed consent before participating in the study. The informed consent form presented two options (I agree/I disagree). Participants who chose the 'I agree' option were allowed to fill the questionnaire, and participants were informed that they could withdraw their participation at any time during the process. College students' participation in the research was voluntary and anonymous. The data analysis has also been guided by the principals of impartiality they are, and aiming to transformation of open and online learning without any preconditions.

## Independent variables

**COVID-19-Related Risk Perception Scale (CRPS).** The CRP level was evaluated by the COVID-19-Related Risk Perception Scale (CRPS), which is based on the SARS Risk Perception Questionnaire (SRPQ) [29]; Chinese SARS Risk Perception Questionnaire (CSPPQ) [30], and Typhoon Controllability and Familiarity Scale (TCFS) [31]. Shi et al. [30] developed the CSPPQ based on Slovic's study [8]. Li [31] developed the TCFS based on the CSPPQ. Thirty participants were involved in a pilot study. To test the CRPS we used the exploratory factor analysis. Along with the preventive measures, the infection of COVID-19, the route of transmission of COVID-19, and the risk of transmission related to the symptoms of COVID-19 were obtained. 'To what extent do you think you are familiar with the symptoms of COVID-19?' and 'To what extent do you think it is possible to control the spread of COVID-19?' are sample questions included in the questionnaire. Confirmatory factor analysis (CFA) of the formal CRPS showed that the scale had good structural validity: $\chi^2/df = 3.86$, CFI (comparative fit index) = 0.90, RMSEA (root–mean–square error of approximation) = 0.066, SMR (standardized root–mean–square residual) = 0.048. The results of CRP showed a Cronbach's alpha of

0.76 (pilot study) and 0.86 (main study). Responses were given on a 5-point Likert-type scale (0 = not at all to 5 = absolutely yes).

Positive and Negative Affect Scale (PANAS-Revision). Watson et al. [32] developed the Positive and Negative Affect Scale (PANAS) based on a two-dimensional structure of emotion. Huang et al. [33] developed the Chinese version of the PANAS in 2003 and showed good reliability and validity in the community. Responses were given on a 5-point Likert-type scale (0 = not at all to 5 = absolutely yes).

Following the introductions, all the participants finished an emotional report form [34], where they rated themselves on a 4-point Likert-type scale (1 = not at all to 4 = extremely) by evaluating the degree to which they experienced particular emotions (amusement, anger, anxiety, disgust, energy, industriousness, enthusiasm, determination, attentiveness, dynamism, restlessness, hostility, irritability, nervousness, fidgeting, fearfulness, engagement, joy, interest, and sadness). We used CFA to test PE and NE. In the pilot study, PE had a Cronbach's alpha of 0.87, and NE had a Cronbach's alpha = 0.76. The composite formal measures of PE found Cronbach's alpha of 0.90, and for NE, 0.89.

## Dependent variables

**Coping Response of COVID-19 Scale (CRCS).** We evaluated the CRC using a 12-item scale based on the Chinese Coping Response to SARS Scale (CCRSS) [10] and Chinese Coping Behavior Scale (CCBS) [31]. Shi et al. [10] developed the CSPPQ based on Billings and Moos' [1] study [13]. Li's [31] CCBS translated and adapted Moos' Coping Response Inventory (CRI) [35]. In the CRCS, the items were grouped into three categories of coping response: active coping behaviour items, self-protection items from the CCRSS, and risk-taking behaviour items from the CCBS. Regarding active coping behaviour, a sample item is 'You take the initiative to reduce the frequency of going out; for self-protection behaviour, 'Do you exercise every day?' and for risk-taking behaviour, 'You go out without a mask'. CFA showed that the formal scale had good structural validity: $\chi^2$/df = 5.23, CFI = 0.90, RMSEA = 0.067, SRMR = 0.035. The reliability of this scale was measured. The results of CRC showed a Cronbach's alpha of 0.78 (pilot study) and 0.84 (main study).

## Data analysis

First, we computed the descriptive statistics and correlation coefficients for the data collected. Second, we specified and tested six mediation models for each of the three categories of CRC. In these models, the mediating hypothesis was tested by regressing CRP on PE and NE, which were regressed on ARB, SPB, or ATB. Prior to the analyses, all predictor variables (i.e. CRP, PE, and NE) and the dependent variable (i.e. ARB) were standardized. The mediation models were tested using Mplus (v. 8.3). Maximum likelihood estimator (ML) was used to estimate model parameters and bootstrap method (n = 1000) was implemented to test the statistical significance of the indirect and the bias-corrected bootstrapped standard errors. While examining the 95% bias-corrected bootstrap confidence intervals [CIs] around the standardised indirect correlations, the results regarding the mediating effects were based on whether the indirect pathways were statistically significant. The model fit was measured using the RMSEA, CFI, SRMR, the Tucker–Lewis index (TLI), and $\chi^2$/df values.

## Results

Current study was cross-sectional, collected the data online, and comprised 1038 valid responses were collected from six universities in three provinces of China. Of these, 762 (73.41%) were females, and 276 (26.59%) were males. Regarding their academic year, 101

**Table 1. Descriptive statistical analysis and correlation analysis results among the variables.**

|  | M | SD | 1 | 2 | 3 | 4 | 5 | 6 |
|---|---|---|---|---|---|---|---|---|
| **1. CRP** | 3.162 | 0.810 | - |  |  |  |  |  |
| **2. PE** | 3.221 | 0.803 | .302** | - |  |  |  |  |
| **3. NE** | 2.054 | 0.762 | -.136** | 0.018 | - |  |  |  |
| **4. ARB** | 4.45 | 1.190 | .592** | .330** | -.087** | - |  |  |
| **5. SPB** | 3.534 | 0.503 | .508** | .189** | -.075* | .394** | - |  |
| **6. RTB** | 2.020 | 0.758 | -.105** | 0.009 | .272** | -.101** | -.076* | - |

** p < 0.01 (two-tailed),

* p < 0.05 (two-tailed).

M = Mean, SD = Standard deviation, CRP = COVID-19-related risk perception, PE = Positive emotion, NE = Negative emotion, ARB = Active-response behaviour,
SPB = Self-protection behaviour, RTB = Risk-taking behaviour. n = 1038.

(9.73%) were seniors, 235 (22.64%) were sophomores, 274 (26.40%) were juniors, and 428 (41.23%) were freshmen. Their ages ranged from 17 to 26 years (Mean [M] = 20.5, Standard Deviation [SD] = 2.69). The sample is consistent with the local population distribution characteristics. Further, 20 (1.90%) participants lived in Wuhan (capital of Hubei province), 124 (11.95%) were from other areas in Hubei, and 894 (86.10%) were from various areas in China. Among them, 268 (25.82%) were from COVID-19 high-risk areas (CHRAs), 291 (28.03%) were from COVID-19 medium-risk areas (CMRAs), and 438 (42.20%) were from COVID-19 low-risk areas (CLRAs); one had missing data.

A Pearson correlation test was performed using IBM SPSS 21.0 to check the construct validity of each dimension; good correlations were found, as presented in Table 1.

Significant differences in familiarity of the four types of risk events (F = 0.73, $p < 0.001$) were observed in the variance test, from the perspective of the known risk. The route of transmission, the preventive measures, the infection of COVID-19, and the symptoms of COVID-19, were the four events found from the multiple comparisons of the most known to the most unknown. Significant differences in the degree of control for risk events (F = 13.128, $p < 0.001$) were seen from the perspective of the risk control, in order from controllable to uncontrollable, preventive measures, infection of COVID-19, symptoms of COVID-19, and route of transmission.

Table 2 shows COVID-19-related risk in early March 2020 in China tended to be perceived as known and controllable. However, the transmission route was seen as uncontrollable and unfamiliar; that is to say, the college students felt the most vulnerable concerning route-of-transmission problems with COVID-19. This was followed by the infection of COVID-19, which was familiar and uncontrollable; that is, although people felt certainty regarding the

**Table 2. The results of COVID-19-related risk perception.**

| Categories of Risk | Familiar | | Controllable | |
|---|---|---|---|---|
|  | M | SD | M | SD |
| **Risk transmission in relation to symptoms of COVID-19** | 3.64 | 0.97 | 2.62 | 0.62 |
| **Route of transmission** | 2.58 | 0.48 | 2.38 | 0.61 |
| **Infection of COVID-19** | 3.42 | 0.78 | 2.78 | 0.70 |
| **Preventive measure** | 3.26 | 0.69 | 4.56 | 1.26 |
| **Overall COVID-19-related risk perception** | 3.23 | 0.83 | 3.09 | 0.80 |

M = Mean; SD = Standard deviation.

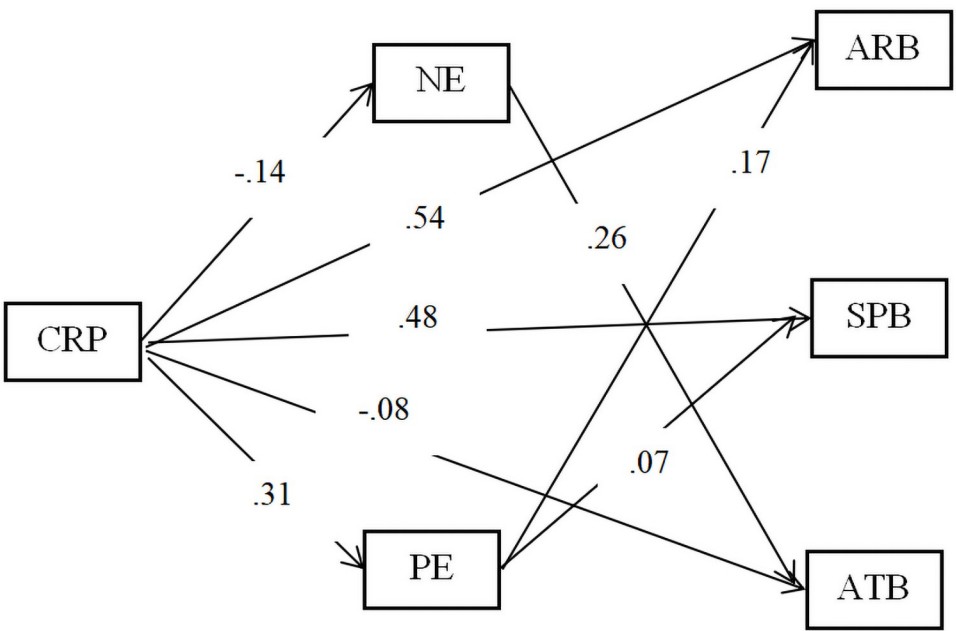

**Fig 2. The mediating effect of PE and NE on the relationship between CRP and CRC.** SPB = Self-protection behaviour, PE = positive emotion, NE = negative emotion, ARB = Active-response behaviour, SPB = Self-protection behaviour, CRP = COVID-19-related risk perception, RTB = Risk-taking behaviour. X = CRP, W1 = NE, W2 = PE, Y1 = ARB, Y2 = SPB, Y3 = RTB. Non-significant coefficients were not presented. Demographic variables controlled; n = 1038.

infection of COVID-19, they still could not control it. Other events (preventive measures) were known and controllable for relatively low risk.

For testing the model of the CRC–CRP relationship mediated by NE and PE, the SEM was adopted (Fig 2). Regarding the RMSEA statistics, CFI, SRMR, (TLI), and the $\chi^2/df$, the fit indices were satisfactory, $\chi^2/df$ = CFI = 0.99, TLI = 0.94. Bootstrapping tests indicated that the indirect influences for the mediating paths were different from zero and were small to medium in magnitude (Table 3). The mediated model included six indirect effects: Ind1: X➔W1➔Y1; Ind2: X➔W2➔Y1; Ind3: X➔W1➔Y2; Ind4: X➔W2➔Y2; Ind5: X➔W1➔Y3; Ind6: X➔W2➔Y3. Among them, three mediating effects were statistically significant: X➔W2➔Y1, X➔W2➔Y2, and X➔W1➔Y3 (Table 3). Hence, our hypotheses were partly supported.

## Discussion

### Perception of familiar and controllable COVID-19-related risk

The respondents reported a greater sense of control over the preventive measures than their sense of control over COVID-19 infection. They perceived the risk events to be uncontrollable despite knowing the symptoms well. Albeit being mediated, the risk perception among the college students would have been further reduced if the Chinese government strengthened the public information efforts on the infectious, curative, and preventive aspects of COVID-19 since March 2020. Finding the COVID-19-related route of transmission problems as being unknown and uncontrollable was a noteworthy issue. People tend to be more afraid of an issue if the uncertainty surrounding a specific risk is high, as confirmed by research on perceptions of risks. Only providing information about the risk could restore hope and improve people's risk assessment [36].

**Table 3. Mediation model: Indirect effect between CRP and CRC through PE and NE.**

| Effect from CRP to ARB | | | |
|---|---|---|---|
| | β | SE | 95%CI |
| Total Effect | 0.60** | 0.02 | [0.53, 0.66] |
| Indirect Effect | 0.05** | 0.01 | [0.03, 0.08] |
| Direct Effect | 0.54** | 0.03 | [0.47, 0.60] |
| Ind1: X➔W1➔Y1 | 0.00 | 0.00 | [-0.01, 0.01] |
| Ind2: X➔W2➔Y1 | 0.05** | 0.01 | [0.03, 0.08] |
| Effect from CRP to SPB | | | |
| Total effect | 0.51** | 0.03 | [0.42, 0.59] |
| Indirect Effect | 0.03* | 0.01 | [0.01, 0.04] |
| Direct Effect | 0.48** | 0.04 | [0.40, 0.58] |
| Ind3: X➔W1➔Y2 | 0.00 | 0.00 | [-0.01, 0.01] |
| Ind4: X➔W2➔Y2 | 0.03* | 0.01 | [0.01, 0.04] |
| Effect from CRP to ATB | | | |
| Total effect | -0.11** | 0.03 | [-0.19, -0.02] |
| Indirect Effect | -0.03* | 0.02 | [-0.05, -0.01] |
| Direct Effect | -0.08** | 0.03 | [-0.16, 0.03] |
| Ind5: X➔W1➔Y3 | -0.04** | 0.01 | [-0.07, -0.02] |
| Ind6: X➔W2➔Y3 | 0.01 | 0.01 | [-0.02, 0.04] |

Bootstrap resample = 1000, β is a standardized coefficient, SE is a Std. Error, and CI is a confidence interval. With respect to effect sizes, standardized indirect effects around 0.01 were interpreted as small, effects ~0.09 were considered medium, and effects ~0.25 were considered large. PE = positive emotion, NE = negative emotion, ARB = Active-response behaviour, SPB = Self-protection behaviour, CRP = COVID-19-related risk perception, RTB = Risk-taking behaviour. X = CRP, W1 = NE, W2 = PE, Y1 = ARB, Y2 = SPB, Y3 = RTB, n = 1038,
**p < 0.01,
*p < 0.05 (two-tailed).

## The relationship between COVID-19-related risk perception and coping response to COVID-19

In the general analysis based on structured modelling, three categories of CRC among college students were significantly correlated with CRP, namely risk-taking behaviour, self-protection behaviour, and active coping behaviour. This finding aligned with the research results indicating that the perception of risk related to terrorism differed by the type of coping to them. Regarding coping behaviour, risk perception is a crucial predictive variable [5,8]. Further, in predicting adaptive behaviours, the perception of health risk could be considered a critical psychological element, as stated by Ye et al. [37]. People with a higher level of risk perception are inclined to accept information about warnings more easily and earlier, as revealed by earlier studies [37–39].

CRP influences active coping and self-protection behaviours; therefore, numerous scholars reveal that the CRP of college students positively influences their self-protective behaviour, and active coping behaviour has not been surprising as per the relevant research [40]. The impact of risk perceptions on risk-related activities of teenagers was examined, and indices were developed in earlier research [41]. Active response to self-protection and coping behaviour is seen in students with a high-risk perception of the preventive measures of COVID-19, the infection, the route of transmission, and the symptoms of COVID-19, than students with lower levels of CRP. Our findings that greater levels of risk perception were likely to enhance

the willingness of people to deal with environmental risks were consistent with the findings of Dominicis et al. [42].

## PE and NE mediate the relationship between CRP and CRC

Emotion affected coping strategies, supporting many previous researchers [23,24,26,43]. Our hypotheses four to nine were that PE and NE would mediate the relationship between CRP and CRC; our results partly supported these hypotheses. The mediating effect of PE on CRP and positive coping behaviour (ARB and SPB) was significant (Ind2: X➜W2➜Y1, β = 0.05, CI [0.03, 0.08]; Ind4: X➜W2➜Y2,β = 0.03,CI [0.01, 0.04]), but it cannot predict risk-taking behaviour, and there was no mediating effect between CRP and risk-taking behaviour (Ind6: X➜W2➜Y3, β = 0.01, CI [-0.02, 0.04]), while NE could positively predict risk-taking behaviour in public health crisis events (Ind5: X➜W1➜Y3, β = -0.04, CI [-0.07, -0.02]). However, there was no mediating effect of NE on the correlation between CRP and positive coping behaviour (Ind1: X➜W1➜Y1, β = 0.00, CI [-0.01, 0.01]; Ind3: X➜W1➜Y2, β = 0.00, CI [-0.01, 0.01]). The ARB to COVID-19 could be enhanced by CRP as implied. Nonetheless, the coping response of college students to COVID-19 was influenced by the mediation of NE/PE in the relationship with CRP. The degree to which the CRP increased the CRC amongst the students with high PE was superior to those with low PE. Additionally, greater positive coping responses were displayed by students with higher positive emotions than the other study participants. Few studies have explored the influence of interacting and mediating variables on the relationship between the CRP and the CRC, despite several earlier studies on CRP and CRC.

Students from the universities in China had higher levels of PE than their peers besides possessing superior psychological resources and ARB, as revered by our study. Previous related reports showed that more frequent use of emotion-oriented coping at the peak of SARS relieved the sadness and anger that people of all ages experienced throughout the outbreak, and more frequent use of problem-oriented coping relieved adults' sadness [44]. Moreover, students with high levels of PE exhibited engagement in COVID-19 response as they had more psychological capital than others did. However, our findings also indicated that NE significantly mediated CRP and RTB. Research has established a close association between emotion and public coping response [24,43,44]. As participants' emotions cause more engagement in coping with COVID-19, students who perceive that they are exposed to risk events will experience negative emotions [24].

Compared with NE, PE plays a greater mediating role between CRP and college students' coping behaviour to COVID-19. This result is similar to Kong and Zhao's study [45], which found that the mediating effect of positive emotions is greater than that of negative emotions. Researchers believe that this may be because positive emotions are the main source of people's power, which can pre-process individual cognition, emotion, and behaviour to establish and expand individual and social resources [45]. A higher level of CRP may help college students transform NE into PE and increase and maintain the frequency of PE so that they can experience more frequent PE and less frequent NE; consequently, they would demonstrate a higher degree of positive coping behaviours and relatively low RTB. However, in the current study, PE and NE played a partial mediating role between CRP and coping response. PE partially mediated positive coping behaviours (ARB and SPB) of COVID-19, while NE partially mediated CRP and RTB, and the difference was statistically significant. The existing literature and data make it difficult to determine the underlying reasons that necessitate further exploration.

## Conclusion

We posit that PE and NE partially mediated the relationship between risk perception and coping response adopted by Chinese college students in response to the COVID-19 pandemic. We built a model composed of nine hypotheses and partly verified it with a sample of 1038 valid responses. We have confirmed that CRP directly affects the coping behaviour of college students and indirectly affects their coping behaviour toward COVID-19 through PE and NE. Significant CRP effects were found, predicting all three coping strategies of CRC. In analyses, after controlling for demographic variables, the data empirically confirmed that PE partially mediated the relationship between CRP and ARB and between CRP and SPB, while NE partly mediated the relationship between CRP and RTB. It was partly verified that PE and NE played a mediating role in CRP and CRC. These findings contribute to the literature on COVID-19-related risk perception illuminating how NE and PE mediate COVID-19-related risk perception in university students.

Our research is novel overall; with very little information on the relationship between PE/NE and different coping responses to COVID-19 amongst the college students in China, certain studies have focused exclusively on college students. Regarding COVID-19, the PE/NE could have potential associations with specific coping behaviours, as confirmed by our research. Leading further to an improvement in the coping response of the college students to COVID-19, by determining the mode in which the PE and NE mediate the pertinent risk perception, these could be considered crucial contributions to the research field to understand the students and the higher education system. Our study results could help expand the limited literature in terms of the higher education system and students. Hence, this study could enrich the theory of earlier research by providing public health management suggestions.

## Implications

In the face of an emergency such as the COVID-19 pandemic, this study illustrated a theoretical framework for risk perception and its correlation with public responses. However, this relationship alone does not adequately explain which monitoring and warning mechanisms are required for effective communication with the public and for equitable intervention strategies to cope with a public health emergency such as COVID-19.

Along with the awakening of the public's PE, the government should pay attention to the mediating effects of the relationship, the predictive effect of NE on the risk-taking behaviour, and the differentiation of the emotional types regarding public health management. The risk perceptions are impacted by the degree of control and the various factors of information by the government department in public health emergencies. As a group sensitive to information, college students could enlarge their risk perception, leading to rational group behaviour, under the joint action of these factors. Hence, the importance of effective risk communication could hardly be emphasized [8]. Through public health crisis preparation and management, the communication of risk needs to be meticulously calibrated.

Our results suggest that students with high CRP reserves will exhibit increased active coping behaviour, self-protective behaviour, and lower risk-taking behaviour. Moreover, the awareness of the need for independent exercise and establishing the concept of lifelong exercise need to be cultivated in the college students by improving the daily activities for the students like sports competitions and lectures, through important CRP interventions as proposed by Liu et al. [40]. To reduce RTB while improving their self-protective and ARB, we recommend enhancing Chinese college students' CRP. Our findings suggest focusing on CRP, with fewer NE and more PE to reduce the RTB in the Chinese college students and improve ARB.

Therefore, enhancing college students' awareness of the need for independent physical exercise is an important measure for disease prevention.

## Study limitations

The relatively limited role of the mediator variables has been a major limitation of the pertinent study. In the future, more sensitive indicators of mediator variables need to be found, and the development of the measurement tools for coping behaviour needs to be improved with the quest for more sensitive indicators of the mediator variables.

Introducing the issues of subjectivity and predictive validity to a certain extent, the collection of data was self-reported. Integrated assessment mechanisms like interviews to overcome limitations, observations, and teacher ratings would be needed in future research. Besides, sampling was performed regionally, so the regional differences and the small sample size could affect the generalizability of the results, as the sampling was done regionally; hence, a large-scale study needs to be carried out. Additionally, female students accounted for 73.4% of respondents in this study; however, whether gender differences affect the relationship between variables in college students is still unclear; in future research, proportionally, more male students should be invited to participate.

## Author Contributions

**Conceptualization:** Yongtao Gan.

**Data curation:** Qionglin Fu.

**Writing – original draft:** Yongtao Gan.

**Writing – review & editing:** Yongtao Gan.

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
