## [Decision Letter · Decision Letter 0]

13 Sep 2021

PONE-D-21-24157Risk perception and coping response to COVID-19 is mediated by positive and negative emotions: a study on Chinese college studentsPLOS ONE

Dear Dr. Fu,

Thank you for submitting your manuscript to PLOS ONE. After careful consideration, we feel that it has merit but does not fully meet PLOS ONE’s publication criteria as it currently stands. Therefore, we invite you to submit a revised version of the manuscript that addresses the points raised during the review process.

We look forward to receiving your revised manuscript.

Kind regards,

Amir H. Pakpour, Ph.D.

Academic Editor

PLOS ONE

Journal Requirements:

Reviewers' comments:

Reviewer's Responses to Questions

**Comments to the Author**

1. Is the manuscript technically sound, and do the data support the conclusions?

Reviewer #1: Yes

Reviewer #2: Partly

2. Has the statistical analysis been performed appropriately and rigorously? 

Reviewer #1: Yes

Reviewer #2: I Don't Know

3. Have the authors made all data underlying the findings in their manuscript fully available?

Reviewer #1: Yes

Reviewer #2: Yes

4. Is the manuscript presented in an intelligible fashion and written in standard English?

Reviewer #1: Yes

Reviewer #2: Yes

5. Review Comments to the Author

Reviewer #1: Title: Risk perception and coping response to COVID-19 is mediated by positive and negative emotions: a study on Chinese college students

Abstract: Study time, Study measures, Sampling procedure and main statistical tests should be explained in abstract. main numerical findings should be reported in the abstract. Conclusion should be provided based on study findings.

Introduction: Some recent literature can be helpful for authors to enhance the introduction DOI: 10.4103/shb.shb_9_21, DOI: 10.4103/shb.shb_32_21, DOI: 10.4103/shb.shb_1_21, DOI: 10.4103/shb.shb_24_21, 10.1111/jsr.13432, /doi.org/10.1016/j.eclinm.2021.100916. Introduction is too long in current format, please shorten this section as possible.

Methods: Please correct the study design as cross sectional. Please provide the ethics code provided for this study. Please move demographic characteristics of participants to first paragraph of results (detailed be provided in table 1 as demographic characteristics and summary in text). How target universities were selected?

Discussion, conclusion, study implication and limitations are well explained.

Reviewer #2: The study entitled “Risk perception and coping response to COVID-19 is mediated by positive and negative emotions: a study on Chinese college students” used 1038 university students’ data to investigate the relationships between risk perception of COVID-19, coping response to COVID-19, positive emotions, and negative emotions. The authors used regression models and structural equation modeling to test their hypotheses, including the mediated effects of positive and negative emotions. The results indicated that both types of emotions are significant mediators in the association between risk perception to COVID-19 and coping response to COVID-19. Although the study reports some new information to the current literature, the manuscript suffers from some drawbacks and I would encourage the authors substantially revise their work.

1. In general, the readability of the present manuscript is poor. Specifically, typos (e.g., standardized coefficient instead of standard coefficient), unexplained abbreviations (e.g., Table 1) with a lot of abbreviations in the main text, awkward and lengthy sentences (e.g., This research attempted to develop a two-mediator model to comprehend the effect mechanism of COVID-19-related risk perception on coping responses to COVID-19; we recommend that public health managers should consider differences in coping response behaviours and address the different mediating roles of positive and negative emotions in coping with public health emergencies) hugely disrupt one’s reading in the manuscript. The author should consult native English speakers who are experienced in scientific writing to substantially improve the manuscript’s quality.

2. The Introduction should use some systematic reviews to strengthen the issue of psychological distress during the COVID-19 pandemic.

Olashore AA, Akanni OO, Fela-Thomas AL, Khutsafalo K. The psychological impact of COVID-19 on health-care workers in African Countries: A systematic review. Asian J Soc Health Behav 2021;4:85-9

Rajabimajd N, Alimoradi Z, Griffiths MD. Impact of COVID-19-related fear and anxiety on job attributes: A systematic review. Asian J Soc Health Behav 2021;4:51-5

Alimoradi Z, Broström A, Tsang HWH, et al. Sleep problems during COVID-19 pandemic and its' association to psychological distress: A systematic review and meta-analysis. EClinicalMedicine. 2021;36:100916. doi:10.1016/j.eclinm.2021.100916

3. The authors should remove all the publication year when they cite a reference in the main text. For example, it should be Shi et al. observed instead of Shi et al. (2003) observed.

4. There are too many hypotheses in the manuscript, and thus one cannot capture the main purpose of the present study.

5. The authors should have a paragraph to clearly explain their entire model. In the present manuscript, the authors only provide scattered and fragment information for readers. This cannot give the readers a full picture of what the present study was doing.

6. The Results section is not concise and it includes information that should is not belonged in the Results section. For example, (1) the Results section should be citation free; (2) how to construct the stepwise regression should be described in the Data analysis section; (3) how the model tested in the structural equation modeling should also be described in the Data analysis section.

7. I wonder why the authors want to conduct the regression models given that all their hypotheses can be answered by the structural equation model.

8. The authors should provide detailed information in their structural equation modeling. For example, what is the estimator? How did the authors construct the model (did the authors consider latent construct and measurement model in the structural equation modeling)?

9. The study is a cross-sectional design; therefore, the authors should explicitly indicate the causality cannot be concluded using the present study’s findings. Moreover, the authors should be aware of their language use in the present study to avoid implying causality (e.g., while NE could positively predict risk-taking behaviour in public health crisis events).

6. PLOS authors have the option to publish the peer review history of their article (what does this mean?). If published, this will include your full peer review and any attached files.

Reviewer #1: No

Reviewer #2: No

---

## [Author Response · Author response to Decision Letter 0]

20 Nov 2021

Dear Dr. Pakpour:

We appreciate the reviewers’ comments regarding our manuscript “Risk perception and coping response to COVID-19 mediated by positive and negative emotions: a study on Chinese college students.”

We thank the reviewers for their thoughtful and constructive comments and suggestions. Overall, the comments are fair, encouraging, and insightful. We have learned a great deal from them. After carefully studying the reviewers’ comments, we made several substantial changes to the paper to incorporate the recommendations, responding to the feedback as much as possible. The revised parts are highlighted in red font in the new draft. Moreover, various grammar and spelling errors were corrected throughout the paper. We have included a point-by-point response to the reviewers’ comments below.

If you have any further questions about the paper, please do not hesitate to let us know.

Sincerely yours, 

Authors

Reviewers' comments:

Reviewer's Responses to Questions

Comments to the Author

1. Is the manuscript technically sound, and do the data support the conclusions?

Reviewer #1: Yes

Reviewer #2: Partly

2. Has the statistical analysis been performed appropriately and rigorously?

Reviewer #1: Yes

Reviewer #2: I Don't Know

3. Have the authors made all data underlying the findings in their manuscript fully available?

Reviewer #1: Yes

Reviewer #2: Yes

4. Is the manuscript presented in an intelligible fashion and written in standard English?

Reviewer #1: Yes

Reviewer #2: Yes

5. Review Comments to the Author

Reviewer #1: 

We greatly appreciate your careful review of our manuscript and your invaluable suggestions. We have carefully considered each of the comments and revised the manuscript accordingly.

1.Title: Risk perception and coping response to COVID-19 is mediated by positive and negative emotions: a study on Chinese college students

Abstract: Study time, Study measures, Sampling procedure and main statistical tests should be explained in abstract. main numerical findings should be reported in the abstract. Conclusion should be provided based on study findings.

Response: Thank you for pointing this out.We have include the study duration and measures, sampling procedure, and the main statistical tests have been explained in the Abstract. We have also incorporated the numerical findings revised the conclusion as per your suggestion. 

2.Introduction: Some recent literature can be helpful for authors to enhance the introduction DOI: 10.4103/shb.shb_9_21, DOI: 10.4103/shb.shb_32_21, DOI: 10.4103/shb.shb_1_21, DOI: 10.4103/shb.shb_24_21, 10.1111/jsr.13432, /doi.org/10.1016/j.eclinm.2021.100916. 

Introduction is too long in current format, please shorten this section as possible.

Response: Thank you for pointing this out and providing the literature resources. We have downloaded these six papers. We hope our Introduction section is more updated.

3.Methods: Please correct the study design as cross sectional. Please provide the ethics code provided for this study. Please move demographic characteristics of participants to first paragraph of results (detailed be provided in table 1 as demographic characteristics and summary in text). How target universities were selected?

Response: Thank you for pointing this out! We have addressed these issues as follows:

1) We have corrected the study to a cross-sectional design as you have suggested;

2) We have provided the ethics approval number in this study on P7.

“This study was approved by Research Ethics Committee of the Shantou University medical college (SUMC-2020-80).”

3) We have moved demographic characteristics of the participants to the first paragraph of the Results section.

4) First, the universities in the Hubei, Guangdong, and Shandong Provinces were sent an invitation e-mail with the study purpose information and a link to an online survey. Six universities responded and agreed to enrol their students in the study.

4.Discussion, conclusion, study implication and limitations are well explained.

Response: Thank you!

Reviewer #2: 

We greatly appreciate your careful review of our manuscript and your invaluable suggestions. We have carefully considered the comments and revised the manuscript accordingly.

The study entitled “Risk perception and coping response to COVID-19 is mediated by positive and negative emotions: a study on Chinese college students” used 1038 university students’ data to investigate the relationships between risk perception of COVID-19, coping response to COVID-19, positive emotions, and negative emotions. The authors used regression models and structural equation modeling to test their hypotheses, including the mediated effects of positive and negative emotions. The results indicated that both types of emotions are significant mediators in the association between risk perception to COVID-19 and coping response to COVID-19. Although the study reports some new information to the current literature, the manuscript suffers from some drawbacks and I would encourage the authors substantially revise their work.

1. In general, the readability of the present manuscript is poor. Specifically, typos (e.g., standardized coefficient instead of standard coefficient), unexplained abbreviations (e.g., Table 1) with a lot of abbreviations in the main text, awkward and lengthy sentences (e.g., This research attempted to develop a two-mediator model to comprehend the effect mechanism of COVID-19-related risk perception on coping responses to COVID-19; we recommend that public health managers should consider differences in coping response behaviours and address the different mediating roles of positive and negative emotions in coping with public health emergencies) hugely disrupt one’s reading in the manuscript. The author should consult native English speakers who are experienced in scientific writing to substantially improve the manuscript’s quality.

Response: Thank you for pointing this out. We have reorganized the manuscript and submitted it to Editage (www.editage.com.cn) for language editing; we expect that it will therefore be more readable and well-written.

2. The Introduction should use some systematic reviews to strengthen the issue of psychological distress during the COVID-19 pandemic.

Olashore AA, Akanni OO, Fela-Thomas AL, Khutsafalo K. The psychological impact of COVID-19 on health-care workers in African Countries: A systematic review. Asian J Soc Health Behav 2021;4:85-9

Rajabimajd N, Alimoradi Z, Griffiths MD. Impact of COVID-19-related fear and anxiety on job attributes: A systematic review. Asian J Soc Health Behav 2021;4:51-5

Alimoradi Z, Broström A, Tsang HWH, et al. Sleep problems during COVID-19 pandemic and its' association to psychological distress: A systematic review and meta-analysis. EClinicalMedicine. 2021;36:100916. doi:10.1016/j.eclinm.2021.100916

Response: Thank you for pointing this out and providing the literature resources. We have downloaded these three papers. We hope our Introduction section is more updated.

3. The authors should remove all the publication year when they cite a reference in the main text. For example, it should be Shi et al. observed instead of Shi et al. (2003) observed.

Response: Thank you for pointing this out. We have revised this as per your suggestion .Although publication years have been removed, the citation numbers have been placed next to the author names as per the Vancouver format recommended by the journal.

4. There are too many hypotheses in the manuscript, and thus one cannot capture the main purpose of the present study.

Response: Thank you for pointing this out. We have revised the hypotheses on P5 as follows:

Therefore, in accordance with previous research, CRP may also predict PE, NE, and CRC. Thus, this study hypothesized that PE and NE would mediate the associations between CRP and three categories of CRC (i.e., ARB, SPB, and ATB).

5. The authors should have a paragraph to clearly explain their entire model. In the present manuscript, the authors only provide scattered and fragment information for readers. This cannot give the readers a full picture of what the present study was doing.

Response: Thank you for pointing this out. We have addressed this issue as follows:

(1)We have explained the entire model in the Data Analysis section on P9: “Second, we specified and tested six mediation models for each of the three categories of CRC. In these models, the mediating hypothesis was tested by regressing CRP on PE and NE, which were regressed on ARB, SPB, or ATB. Prior to the analyses, all predictor variables (i.e., CRP, PE, and NE) and the dependent variable (i.e., ARB) were standardized.”

(2)We modified the Independent variables and Dependent variables section in the “Materials and Methods” section on P7-8

(3)We have added the concept model (Figure 1)on P5

6. The Results section is not concise and it includes information that should is not belonged in the Results section. For example, (1) the Results section should be citation free [2]; how to construct the stepwise regression should be described in the Data analysis section [3]; how the model tested in the structural equation modeling should also be described in the Data analysis section.

Response: Thank you for pointing this out.We have addressed this issue as follows: 

(1)We have deleted the citations in the Results section; 

(2)In accordance with your comment 7, we have deleted the stepwise regression model description; 

(3)We have added the description of the structural equation model in the Data analysis section on P9 as follows:

“Second, we specified and tested six mediation models for each of the three categories of CRC. In these models, the mediating hypothesis was tested by regressing CRP on PE and NE, which were regressed on ARB, SPB, or ATB. Prior to the analyses, all predictor variables (i.e., CRP, PE, and NE) and the dependent variable (i.e., ARB) were standardized. The mediation models were tested using Mplus (v. 8.3). Maximum likelihood estimator (ML) was used to estimate model parameters and bootstrap method (n = 1000) was implemented to test the statistical significance of the indirect and the bias-corrected bootstrapped standard errors. While examining the 95% bias-corrected bootstrap confidence intervals [CIs] around the standardised indirect correlations, the results regarding the mediating effects were based on whether the indirect pathways were statistically significant. The model fit was measured using the RMSEA, CFI, SRMR, the Tucker–Lewis index (TLI), and χ2/df values.”.

7. I wonder why the authors want to conduct the regression models given that all their hypotheses can be answered by the structural equation model.

Response: Thank you for pointing this out. We wished to provide further information relevant to our study; thus, we computed and analyzed the regression models. However, you are correct, so we have deleted the regression models and retained only the structural equation model to make the manuscript more concise. 

8. The authors should provide detailed information in their structural equation modeling. For example, what is the estimator? How did the authors construct the model (did the authors consider latent construct and measurement model in the structural equation modeling)?

Response: Thank you for pointing this out. We have addressed this issue as follows:

(1) We have added the structural equation estimator information in the Data analysis section on P9.

(2) We have not considered the latent constructs in the structural equation modeling as this model involves too many variables, but we have provided the fit indices of measurement model in the Materials and Methods section (Independent variables and Dependent variable) and the Results section.

9. The study is a cross-sectional design; therefore, the authors should explicitly indicate the causality cannot be concluded using the present study’s findings. Moreover, the authors should be aware of their language use in the present study to avoid implying causality (e.g., while NE could positively predict risk-taking behaviour in public health crisis events).

Response: Thank you for pointing this out.We have revised the manuscript to incorporate your suggestion.

---

## [Editor Report · Decision Letter 1]

19 Dec 2021

Risk perception and coping response to COVID-19 is mediated by positive and negative emotions: a study on Chinese college students

PONE-D-21-24157R1

Dear Dr. Fu,

We’re pleased to inform you that your manuscript has been judged scientifically suitable for publication and will be formally accepted for publication once it meets all outstanding technical requirements.

Kind regards,

Amir H. Pakpour, Ph.D.

Academic Editor

PLOS ONE
---

## [Editor Report · Acceptance letter]

24 Dec 2021

PONE-D-21-24157R1 

Risk perception and coping response to COVID-19 mediated by positive and negative emotions: a study on Chinese college students 

Dear Dr. Fu:

I'm pleased to inform you that your manuscript has been deemed suitable for publication in PLOS ONE. Congratulations! Your manuscript is now with our production department. 

Kind regards, 

on behalf of

Dr. Amir H. Pakpour 

Academic Editor

PLOS ONE